# Effectiveness of Video Games as Physical Treatment in Patients with Cystic Fibrosis: Systematic Review

**DOI:** 10.3390/s22051902

**Published:** 2022-02-28

**Authors:** Remedios López-Liria, Daniel Checa-Mayordomo, Francisco Antonio Vega-Ramírez, Amelia Victoria García-Luengo, María Ángeles Valverde-Martínez, Patricia Rocamora-Pérez

**Affiliations:** 1Health Research Centre, Department of Nursing, Physiotherapy and Medicine, University of Almería, Carretera del Sacramento s/n, La Cañada de San Urbano, 04120 Almería, Spain; mvm637@ual.es; 2Hum-498 Research Team, University of Almeria, 04120 Almeria, Spain; dany.cheka.m@gmail.com; 3Torrecardenas Hospital, 04009 Almería, Spain; francisco.vega.ramirez.sspa@juntadeandalucia.es; 4FQM228-Research Team, Department of Mathematics, Carretera del Sacramento s/n, La Cañada de San Urbano, 04120 Almería, Spain

**Keywords:** cystic fibrosis, video game, gaming console, physiotherapy, pulmonary rehabilitation

## Abstract

Physical training at home by making individuals play active video games is a new therapeutic strategy to improve the condition of patients with cystic fibrosis (CF). We reviewed studies on the use of video games and their benefits in the treatment of CF. We conducted a systematic review with data from six databases (PubMed, Medline, Scopus, Web of Science, PEDro, and Cochrane library plus) since 2010, according to PRISMA standards. The descriptors were: “Cystic Fibrosis”, “Video Game”, “Gaming Console”, “Pulmonary Rehabilitation”, “Physiotherapy”, and “Physical Therapy”. Nine articles with 320 participants met the inclusion criteria and the study objective. Patients who played active video games showed a high intensity of exercise and higher ventilatory and aerobic capacity compared to the values of these parameters in tests such as the cardiopulmonary stress test or the six-minute walk test. Adequate values of metabolic demand in these patients were recorded after playing certain video games. A high level of treatment adherence and satisfaction was observed in both children and adults. Although the quality of the included studies was moderate, the evidence to confirm these results was insufficient. More robust studies are needed, including those on evaluation and health economics, to determine the effectiveness of the treatment.

## 1. Introduction

Cystic fibrosis (CF) is also known as mucoviscidosis due to the production of copious amounts of sticky mucus [1] that accumulates mainly in the lungs and other organs. It is an autosomal recessive inherited genetic disease caused by a mutation of the CFTR gene (CF Transmembrane Conductance Regulator) and exhibits chronic and fatal pathology due to bronchiectasis and progressive and obstructive pulmonary deterioration [2]. CF is highly prevalent in Europe, Australia, and North America, and is very common among Caucasians compared to its prevalence in other races [2,3]. Its incidence is variable; one in 3000 or 8000 children are born with the disease, and one in 25 are carriers of the defective gene [4].

The symptoms in children at the pulmonary level include chronic coughing or expectoration, infection of the bronchi by bacteria, bronchiectasis (very common in most people with CF), atelectasis, airway obstructions, and nasal polyps. At the gastrointestinal level, the symptoms include rectal prolapses, pancreatitis, biliary cirrhosis, malnutrition, and syndromes caused by the salt loss [5]. Adults might have lung infections, pancreatitis, biliary cirrhosis, and chronic cough, as well as biliary lithiasis, jaundice, sinusitis, and infertility [5].

In CF patients, more changes are observed at the pulmonary level leading to chronic inflammatory diseases of the respiratory tract characterized by pulmonary infections, generally due to the colonization of *Pseudomonas aeruginosa*, *Staphylococcus aureus*, *Haemophilus influenzae*, or *Stenotrophomonas maltophilia* [6]. CF is multisystemic and life-threatening, although the advancement in treatment and care have increased life expectancy to 40–50 years [7,8]. 

For cleaning or clearing secretions, aerosol therapy is recommended with compounds such as hypertonic saline before respiratory physiotherapy sessions, as they reduce exacerbations and the thickness of the mucus, making it easier to expel [9]. Physical exercise prevents disease progression effectively by improving aerobic capacity and lung function [10]. Supervised home training, as part of the treatment, can motivate the patient and improve their physical condition substantially. Active video games are used as an alternative for patients to perform physical training at home [11]. These are widely used in children with chronic pathologies and provide multiple benefits, including reducing sedentarism and preventing long-term health complications, among others [12].

Video games are widely used in respiratory diseases, and, in children with asthma, performing aerobic exercise reduces lung inflammation and increases lung capacity [13]. Although research on CF and video games is limited, some studies have suggested that playing videos games might enhance the strength of patients and increase their exercise tolerance; thus, influencing their quality of life and increasing their adherence to home treatment [14,15]. Although there is some preliminary evidence on the efficacy of playing games for rehabilitation under respiratory conditions, more studies are required to elucidate alternative low-cost exercises (i.e., exergames) for patients with CF to improve their physical condition and pulmonary function and to prevent complications [16]. To our knowledge, no previous systematic review has evaluated the effectiveness of video games on CF. 

In addition, the authors provided an alternative method of physical training for these patients at home. It is safe to say that performing physical activity is more important now than ever for these patients, given their higher risk of being affected by COVID. This more motivating approach can encourage patients to perform and engage in physical activities during isolation for maintaining and/or improving their health.

In this review, we described the latest studies related to video game therapy in CF patients and evaluated its benefits compared to those of other treatments or standard care.

## 2. Materials and Methods

A systematic review of the articles published in the last 10 years (until August 2021) in PubMed, Medline, Scopus, the Web of Science, PEDro, and the Cochrane library plus, according to PRISMA standards [17] and registered in the International prospective register of systematic reviews in the PROSPERO database (CRD42021247065), was performed. 

### 2.1. Search Methods for Identification of Studies 

The following inclusion criteria were used to select the articles between 2010 and 2021: randomized controlled trials (RCTs) and pilot clinical trials or observational studies in any language, without restrictions on the state of publication. Descriptors in English: “Cystic Fibrosis”, “Video Game”, “Gaming Console”, “Pulmonary Rehabilitation”, “Physiotherapy”, and “Physical Therapy”. 

The exclusion criteria were as follows: articles in which video games were applied to another pathology, not addressing CF, or that were not related to the aim in this review; and bibliographical, systematic, or meta-analysis reviews, and doctoral theses. 

The following PICOS eligibility criteria were used for the selection of the articles (participants, intervention, comparator, outcomes, and study design):

Participants were patients with a confirmed diagnosis of CF; only physiotherapy techniques or rehabilitation with video games (of any kind) was used for intervention; the comparator included studies comparing other interventions or no intervention (control group); the outcomes were defined as any objective measure of health, and all clinical data were measured by a validated tool. The study type could be RCTs, pilot trials (with results), or observational studies. 

### 2.2. Data Collection and Analysis

A total of 809 articles were selected based on the abovementioned search strategy. They were analyzed based on the title and the abstract, and 724 articles were excluded as they did not meet the inclusion criteria (Figure 1).

After preliminary screening of the studies that were considered to be potentially relevant (32 articles), two researchers (DCM and RLL) independently read the articles thoroughly and discarded those studies that were duplicates or did not meet the criteria. They consulted a third researcher (MAVM) when there was any disagreement. Finally, nine articles met the inclusion and exclusion criteria and matched the proposed objective of this review. The quality of the randomized controlled trials (RCTs) was assessed using the PEDro scale [18].

We could not conduct a meta-analysis because of the differences in the study design, lack of a control or placebo group, and inconsistencies in the outcomes reported.

## 3. Results

The search strategy in the different selected databases is described in Table 1.

Of the nine articles, 33% were based on clinical trials [19,20,21], 44% were observational studies [22,23,24,25], and the other 22% corresponded to pilot studies [26,27]. There was a total of 375 participants, of whom 320 were CF patients and 55 belonged to the healthy control groups.

The main variables of the articles included in this review are summarized in Table 2.

Additionally, an analysis of the contents of the articles was performed on the following variables.

### 3.1. Evaluation and Questionnaires Used

In most of the selected studies, anthropometric measurements, such as weight, height, and lung function, were initially recorded by spirometry [19,22,23,24,25]. The six-minute walk test (6 MWT) [19,23,24] was conducted in three studies, while the cardiopulmonary exercise test (CPET) was performed in four studies [20,21,22,24]. Additionally, other tests were conducted, such as the modified shuttle test (MSWT) specifically for children with CF, the horizontal jump test (HJT), the medicine ball throwing test (MBTT), and the handgrip test (HGT) with three strength tests [19].

Various scales, such as the OMNI scale [20,22,24] and the ratings of perceived exertion, (RPE) scale [21,25], were used to measure the perceived effort following the tasks performed in the studies. The visual analog scale [20,21] and the modified Borg scale [21,22,23,24,25] were used to measure the level of dyspnea and fatigue.

Additionally, a Likert-type scale was used to assess the level of satisfaction and adherence [19,20,22]. The data obtained showed that most of the subjects were highly satisfied with and adhered to the treatment with video games.

In four articles, a questionnaire on physical activity was used to assess whether participants had previously exercised [19,22,24]; a 20-min structured questionnaire was also used [27]. One study used three Spanish adaptations of the CF revised questionnaire (CFQ-R) [19], which was specific for the patients. For healthy subjects, a questionnaire regarding their respiratory health was also implemented in which questions were asked to determine whether they had asthma, bronchitis, or a family history of such issues [22]. After treatment with video games, parameters, including enjoyment, acceptance, comfort, and the desire to continue, were assessed through written questionnaires (on the acceptability of active video games [19]) or structured interviews [26].

### 3.2. Interventions

The most recent study published [22] was conducted using active video games with Xbox One and Nintendo Wii video game consoles. The researchers recruited healthy adolescents and patients with CF. During their first visit, the forced vital capacity (FCV), expiratory volume in the first second (FEV1), FEV1/FVC, oxygen consumption (VO2), lung ventilation per minute (VE), and oxygen saturation (SpO2) were evaluated through spirometry and CPET, where the participants were instructed to reach their exhaustion limit on a treadmill. In the second meeting, to compare the physiological responses with CPET, a gas mask was used on the participants who played the video games “Just Dance 2015” containing three games (“Love Me Again-John Newman”, “Summer-Calvin Harris”, and “Happy-Pharrell Williams”) on the Xbox One console and “Wii Fit Plus” with three other games (“Obstacle Course”, “Rhythm Boxing”, and “Free Run”) on the Nintendo Wii console. Each game was played for 10 min and was randomly selected for each participant. The results of the video games were similar in both groups, but the participants were considerably less active while playing the four games than during the CPET. While playing the games “Summer” and “Free Run”, a moderate-intensity metabolic demand (>3 METs) (High metabolic demands) was recorded, similar to that of the CPET; however, the video games were more satisfying for all subjects [22]. The study concluded that video games generated a cardiorespiratory response similar to the anaerobic threshold levels found during CPET, indicating that playing video games might be an alternative to exercising for individuals with CF.

A clinical trial (2018) [19] was conducted where children with cystic fibrosis played active video games at home. The researchers evaluated the results of the 6 MWT, MSWT, MBTT, HGT, and HJT. The children played a video game called “EA Sports” on the Nintendo Wii for 30-60 min (five days a week) for six weeks. This game included different mini-games associated with sports, where the children could run, bend over, and exercise their arm muscles. The game could analyze the heart rate with a monitor, and a physiotherapist was appointed to increase patient adherence every week. High metabolic demands (METs) were achieved, which increased gradually, and muscle and cardio-respiratory strength increased after 12 months of follow-up; moreover, great treatment adherence was recorded [19].

In another clinical trial, children with cystic fibrosis were randomly assigned to two groups that later crossed over to compare the results when using the Xbox Kinect console and an exercise bike [20]. In a mini-game called “Rush River” from the video game “Kinect Adventures”, the children controlled a virtual raft down a river with their bodies and had to jump over obstacles and move sideways to collect virtual coins. The game had three levels of intensity that lasted 6 min with two breaks of 1 min each; the session was 20 min long. The exercises proposed with the exercise bike were based on maintaining a constant pedaling rhythm, with changes in resistance, for 20 min. Both tests were supervised by a physiotherapist, who monitored the heart rate and SpO2. The visual analog scale for dyspnea and the OMNI for fatigue were also applied. The results obtained were similar in both tests, except for the perception of fatigue and dyspnea, where the values were lower on the Xbox Kinect than on the exercise bike [20].

To determine the physiological response of CF patients while playing on the Nintendo Wii, a study was conducted where patients played three video games or performed the 6 MWT [19,23,24]. Two walking tests were conducted for 6 min, with a break of 1 h between the two tests. The participants had to walk as fast as possible to record the values for the longest distance traveled. All the video games were 5 min long and were supervised by a physiotherapist. The “Fit Plus” game was played and consisted of aerobic exercises on the “Wii-Fit” platform for balance and coordination with arms and legs. The game “EA Sports Active” was played with various sports-associated mini-games on the “Wii-Active” platform to improve endurance, flexibility, and muscle strength. The game “Family Trainer Extreme Challenge” was played on the “Wii-Fit Trainer” platform with extreme sports to train the whole body in which the child had to jump, move, and avoid obstacles that detected the pressure changes during jumps. The results indicated that greater physiological activity was performed while playing the video games than while performing the 6 MWT, except for the “Wii-Fit” platform [23].

In one of the studies analyzed [24], 60 children were recruited, half of whom had cystic fibrosis, while the other half comprised healthy participants. A spirometry test was performed a month before the test to compare the data obtained later. To perform the test with the active video games, the 6 MWT was performed first. Then, after lying down for 15 min to rest, the participants were randomly selected to play two video games for 15 min with a 5-min break between each session. The games played were “Wii Sports Boxing” and “Wii Fit Free Jogging”. Both games were played with an oximeter on the finger, and the number of steps was monitored with a pedometer worn on the ankle. The results between the two groups were similar for their performance in the video games, but in the 6 MWT, the healthy children covered a greater distance. Additionally, the participants showed higher activity while playing “Wii Fit Free Jogging” than while playing “Wii Sports Boxing”, which did not exceed 3 METs [24].

To determine the intensity of exercise that the Xbox One could achieve, a study was conducted with adult CF patients during two 60-min sessions [25]. The first session involved the CPET following spirometry measurements. The CPET consisted of pedaling for 15 min on an exercise bike while the gases in the blood and the heart rate were monitored. In the second session, the heart rate and SpO2 were measured while working out using the video game “Your Shape Fitness Evolved” on the Xbox Kinect console, where various aerobic activities controlled by a virtual trainer could be performed by punching and jumping with different games and levels of difficulty, such as boxing, dancing, or performing yoga. All participants were allowed to rest until they could continue if symptoms of dyspnea or fatigue were high while exercising. The results of the CPET showed higher values for the heart rate and dyspnea (on the Borg scale) and lower limb fatigue (on the perceived effort scale; RPE). Oxygen desaturation was higher in the CPET than after playing on the Xbox [25].

Two computer video games were designed for studying children with CF while they were connected to a spirometer [27]. The participants were randomly assigned to one group where they underwent inhalation and exhalation tests without feedback from a control software. In the other group, the participants played video games designed in two dimensions and color with feedback. Later, the treatment for the participants was swapped. The tests were conducted daily for 15 min over two or three weeks, where the heart rate and FEV1 were measured. The “Ludicross” game involved forced expirations with the spirometer to clean and fill a water tank in a race car. In the “Creep Frontier” video game, a helicopter was operated to free animals trapped in the mud, shooting air bubbles when blowing through the spirometer. During the gaming periods, there were improvements in the heart rate and FEV1 relative to the control periods [27].

In hospitalized adult patients with CF, a randomized crossover trial was conducted using the Nintendo Wii console compared to an exercise bike and a treadmill [20]. The game “EA Sports” was played with the “Wii-Active” platform in which various sports-related mini-games, such as running, dancing, and boxing, were performed. The interventions lasted for 15 min, with a warm-up and final stretching, supervised by a physiotherapist. The heart rate, dyspnea, and fatigue were similar in all groups, although participants were considerably more satisfied after playing on the Nintendo Wii than after using the exercise bike or treadmill [21].

In another article [26], children hospitalized with CF were studied using a computer game with feedback connected to a digital spirometer. The game involved keeping a green ball on a path for 20 s—after inspiration, the ball went down, and after expiration, the ball went up. Participants gained points if the ball was within the limits of the path and lost them if it went out. They could compete against themselves with great respiratory and visual feedback. The six sessions were 15 min long. The results of coordination, dyspnea, and fatigue were positive. Additionally, the participants were highly interested in using the spirometer with video games [26].

### 3.3. Methodological Quality of the Articles Included

To assess the methodological quality of the articles included, the PEDro scale [18] was used, and the results are displayed in Table 3. 

Eleven items with scores from 0 to 10 points were evaluated. A clinical trial was considered to be rigorous when its score was greater than or equal to 7 points and of low quality when the score was below 6 points.

The scores of the analyzed studies indicated that the studies had a moderate to rigorous methodological quality, except the pilot studies, which were of low quality [26,27]. The methodological quality of the observational studies could not be evaluated [22,23,24,25] using the PEDro scale since the scale is specifically for clinical trials [18].

## 4. Discussion

This systematic review compared the results of studies on the effect of video games on people with CF over the last 10 years by evaluating the physiological benefits and treatment adherence. These trials confirmed, with moderate to rigorous methodological quality, that video games are a potential alternative to physical training methods for rehabilitation and can generate a heart rate response similar to that recorded during conventional exercises. Video games showed greater engagement in activities of sufficient physical intensity, as patients with CF preferred active games to physical training. 

The video game consoles most commonly used were the Nintendo Wii and the Xbox Kinect, except for two clinical trials, where a computer was used [26,27]. No studies have been conducted with other types of consoles, such as the PlayStation, which have also been used in other studies involving children performing physical activities [28].

Computers connected to a spirometer can also be used for these types of patients, not only at a physical level, but also at a respiratory level. In a study by Bingham et al. [26], children not only showed improvements in heart rate, but also in FEV1. In another study, an increase in coordination and a decrease in dyspnea and fatigue were observed [27].

Spirometry was used in almost all the studies to evaluate pulmonary function while performing different activities in video games [19,22,23,24,25]. The cardiovascular responses obtained in the various studies using active video game training modes [19,25] showed that high energy demands similar to those usually recorded during physical exercise could be achieved [28].

The CPET is a widely used method. To compare the results of children and adolescents who played on the Xbox Kinect, Salonini et al. [20] used an exercise bike for the control group; Holmes et al. [25] used the same procedure with adult patients. However, instead of using an exercise bike, Campos et al. [22] used a treadmill for the analysis. Kuys et al. [21] used the exercise bike and the treadmill. 

O’Donovan et al. [24] used the six-minute walk test; del Corral used the same test in their two studies [19,23], but in one of the studies, they added other tests, such as the modified shuttle test (MSWT), the horizontal jump test (HJT), the medicine ball throwing test (MBTT), and the handgrip test (HGT) [19].

In four studies, the normal maximum heart rate levels were recorded after physical exertion by playing video games [21,23,24,25]. However, in the study by Salonini et al. [20], only 40% of the participants using the Xbox Kinect achieved this. In two studies [23,24], the VO2 value was higher after the video game session than that after the six-minute walk test; in another study [25], higher VO2 values were recorded after playing video games than that after the CPET.

Performing aerobic exercise benefits CF patients and healthy people [21,22]. Thus, active video games might be a good alternative to physical training at home for these patients [19,23]. A systematic review analyzed if active video games might be a more engaging alternative to traditional forms of exercise for respiratory patients [29]. These interventions, undertaken for several weeks, can provide similar or greater improvements in the capacity to exercise and other outcomes than traditional exercise, besides being more enjoyable for the subjects. 

A key factor in promoting adherence and motivation was the supervision of physiotherapists while the games were being played by both children [20,23] and adults [21], although in some cases, the physiotherapist only communicated over the telephone every week to motivate the participants [19]. In other cases, the control was conducted by the video game console, where the data of the children monitored were obtained through a virtual trainer incorporated into the videogame [19,23,25]. In the absence of supervision, the capacity to record adverse events might be limited.

The benefits of extending the duration of video games and the number of sessions need to be determined since, in most studies, the duration of each session was very short. For example, in the study by Campos et al. [22], a session was 10 min long, and in the longest study, it lasted 30–60 min [19]. Furthermore, in some studies, training was conducted for only one session [20,21,22,23,24].

In some active video games, the values for the intensity of the activity recorded were lower than those obtained in real practice. This was the case with games that involved boxing [24], obstacle courses, or mini-dance games, such as “Love Me Again” or “Happy”, included in the game “Just Dance 2015”. However, the mini-game “Summer” might be recommended for CF patients, as it recorded values above 3 METs [22], as did the video games involving jogging [24] or running freely [22], where a moderate intensity of aerobic exercise was recorded. The video game “Fitness Evolved”, including boxing and dancing, showed an increase in the intensity of workload among the participants [25].

To achieve a level of physical exertion similar to those of respiratory rehabilitation, selecting the correct active video games is necessary, accounting for the specific condition of each patient. For example, patients with acute exacerbations should perform work of lower intensities for more beneficial effects on lung function [30]. This would involve selecting the most suitable games for each patient.

A fundamental aspect of promoting adherence is that the subjective perception of the patients during training should be positive, allowing them to enjoy themselves while exerting themselves physically. Studies that evaluated the degree of satisfaction found that this was higher when an active video game was played, compared to routine clinical tests, such as the CPET [19,20] or 6 MWT [22]. Similarly, patients reported that they enjoyed playing on the Nintendo Wii more [21].

Inclusion of immediate feedback measures should be considered, such as (a) using techniques for breathing and cleaning the bronchial tree and (b) monitoring treatment programs that use aerobic exercise, as they facilitate greater adherence. This adherence might be achieved because, while playing active video games, the participants can gradually improve their results and monitor their scores and performance [21]. Similarly, in video games, biofeedback is associated with the drainage of the bronchi through a spirometer. Feedback and self-competitive factors provide the setting for patients to progressively become more aware of their breathing [26,27], both in hospitalized children and adults [21,26,27].

Another factor that favors motivation and adherence to treatment is the novelty of the activity. Extending the play-time of the same game is not suggested, as it could be monotonous and might cause the patients to abandon the training method [22].

Investigating the effects associated with competition against other users with the same pathology, or winning prizes, for treatment adherence at home [19] would be interesting in the long term. Competition, besides promoting physiological improvement, could favor socializing among people with similar problems and difficulties, which might provide great psychological support to better cope with the disease [31].

A systematic study investigated the use of video games in the treatment of obstructive respiratory diseases [32] and showed that they were a very useful complementary therapy (enhance rehabilitation programs, improve exercise capacity, muscle strength, and quality of life, as well as the severity, control, and knowledge of the disease). Nine RCTs were included; seven studies were conducted with asthma patients, one including CF patients [19], and one including patients with COPD and other obstructive respiratory diseases. In three of the studies, video games were used as physical training methods, while in the other six articles, video games were used to educate the patients about the disease and the use of medication. A previous meta-analysis showed that educational video games can improve the knowledge and self-management of young people with chronic conditions effectively [33]. Carbonera et al. [15] concluded in their review that interactive video games generate a heart rate intensity similar to that required for physical training in CF patients.

Strategies to address health emergencies and social isolation are necessary, such as those currently being implemented. At the beginning of the COVID-19 pandemic in 2020, there were no data on the prevalence and incidence of infections among these patients [34], but later, few cases of CF patients were reported. This might be due to the effort of relatives to isolate and not infect cystic fibrosis patients, who are highly vulnerable to suffering serious lung problems, have difficulty obtaining lung transplants, and might die due to problems caused by SARS-CoV-2 [35]. Cystic fibrosis centers across Europe canceled routine clinic appointments to prevent unnecessary hospital visits and the spread of the virus. Procedures such as respiratory function testing and bronchoscopy were paused, which negatively affected the well-being of the patients. Phone calls and emails were used to monitor the clinical condition of the patients and to provide psychological support immediately after the lockdown. During long stays in the hospital or at home due to the pandemic, the patients could independently continue treatment using spirometers [26,27] and active video games [19,21,22] to maintain good physical condition. The patients and their families should be provided with tools to support self-monitoring and care management for performing airway clearance and regular exercise during the pandemic [35].

A limitation of this study was that only a few clinical trials used video games for treating CF patients, which prevented us from providing conclusive and rigorous answers about the efficacy and effectiveness of these devices. This limitation could be due to the novelty of the subject matter. The number of participants in the studies was small, as CF is a rare disease. Finally, the type of exercise or test used as the comparator varied among the analyzed studies, and making comparisons was difficult because different video games were played for different durations and purposes. The strengths of this systematic review were the use of the PRISMA guidelines, pre-specified PICOS eligibility criteria, a comprehensive search strategy, and independent screening and data extraction. Considering that this is a relatively new topic in this research area, we identified a limited number of RCTs that met all eligibility criteria and included the description of observational studies with interesting information.

Practical implications. The information provided might facilitate video games to be used as an alternative form of treatment to obtain physiological values similar to the tests that are usually conducted in rehabilitation. We showed the importance of developing alternative procedures that involve greater treatment adherence for chronic diseases, such as cystic fibrosis.

## 5. Conclusions

More studies are needed to assess the effectiveness and long-term adherence to video games in CF patients, as few studies have been conducted in the last 10 years. 

Video games can be a good alternative to conventional treatment, or a very useful complementary therapy, as high metabolic demand and great benefits were recorded at the physiological level after participants played video games. In the studies analyzed, a high level of treatment adherence was recorded when video games were used. Having a playful component made the treatment more entertaining and enhanced the motivation and adherence for continuing the treatment in the medium or long term. However, the mixed results suggested that interventions involving games need to be better designed (i.e., dosage, intensity, and the characteristics of interventions) and thoroughly tested to increase their effect on improving health outcomes.

## Figures and Tables

**Figure 1 sensors-22-01902-f001:**
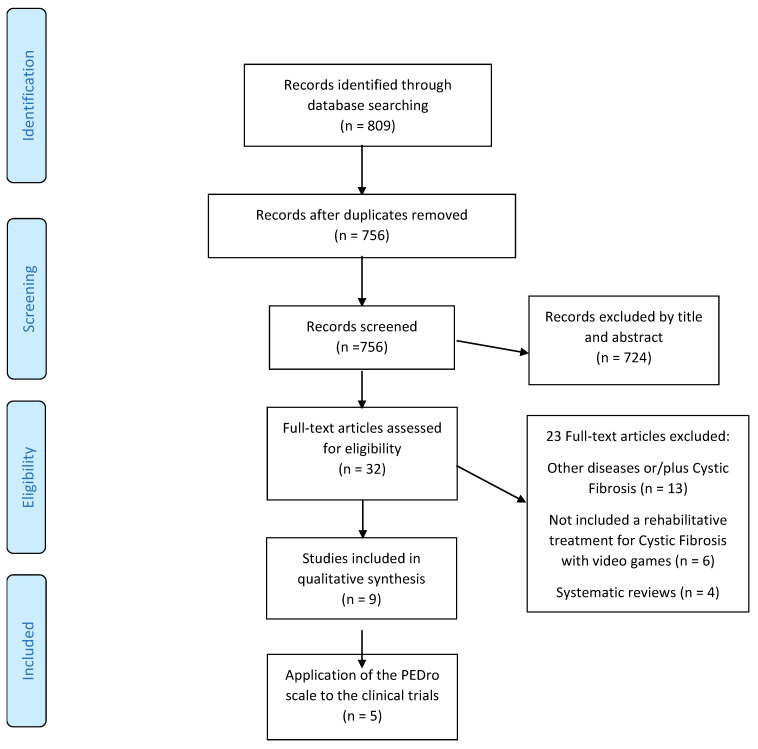
The selected articles from the databases.

**Table 1 sensors-22-01902-t001:** Search strategy in the main databases.

Databases	Descriptors	Results
**PUBMED**	“Pulmonary Rehabilitation” AND “Video Game”	27
	“Cystic Fibrosis” AND “Video Game”	14
	“Cystic Fibrosis” AND “Gaming Console”	4
**MEDLINE**	“Cystic Fibrosis” AND “Gaming Console”	4
	“Cystic Fibrosis” AND “Video Game”	14
**SCOPUS**	“Video Game” AND “Physiotherapy” OR “Physical Therapy”	659
	“Pulmonary Rehabilitation” AND “Video Game” OR “Gaming Console”	17
	“Cystic Fibrosis” AND “Video Game”	19
	“Cystic Fibrosis” AND “Gaming Console”	4
**Web of Science** **(Medline and Scielo)**	“Cystic Fibrosis” AND “Video Game”	25
	“Cystic Fibrosis” AND “Gaming Console”	4
**PEDro**	“Cystic Fibrosis” AND “Video Game”	3
	“Cystic Fibrosis” AND “Gaming Console”	1
**COCHRANE library plus**	“Cystic Fibrosis” AND “Video Game”	14

**Table 2 sensors-22-01902-t002:** The main characteristics of the selected studies.

Author, Year	Type of Study and Participants	Intervention and Follow-Up	Measurement Variables and Results
Campos et al., 2020 [22]	A cross-sectional study was performed: CG = 25 healthy children, EG = 30 children with CF.	The MET was evaluated by playing active videogames on the Nintendo Wii and Xbox One consoles. CG and EG: The participants played six different video games for 10 min.	Measurements were made by spirometry and CPET. The results of the video games were similar to those of the CPET but with a higher level of satisfaction.
Del Corral et al., 2018 [19]	A randomized clinical trial was performed. Forty children and adolescents with CF were randomly assigned to two groups. CG = 20 EG = 20	A home-training program was conducted using video games. CG: No training. EG: Training for five days (30/60 min) for six weeks; follow up of 12 months.	Participants were evaluated using the 6 MWT, MSWT, HJT, MBTT, and HGT. The results were better during post-monitoring testing for the EG.
Salonini et al., 2015 [20]	A randomized crossover trial was conducted. Thirty children with CF were randomly assigned to two groups	One group was assigned to play three video games with the Xbox Kinect (20 min), while the other group was assigned an exercise bike (20 min).	The parameters measured were HR, SpO2, dyspnea, and the perception of fatigue. The group that was assigned the Xbox Kinect performed better than the group that was assigned the bike.
Del Corral et al., 2014 [23]	An observational study was performed with 24 children and adolescents with CF.	The physiological responses while playing on the Nintendo Wii were evaluated. Aerobic exercises were performed while playing three video games for 5 min each.	The parameters VO2, VE, and HR were measured and their values for the video game sessions and the 6 MWT were compared. Higher physiological demands were recorded while playing on the Nintendo Wii.
O’Donovan et al., 2014 [24]	A cross-sectional study was performed: CG = 30 healthy children, EG = 30 children with CF.	The MET was evaluated while playing on the Nintendo Wii console. Aerobic boxing and jogging exercises were performed using two active video games for 15 min.	The parameters VO2, Kcal, and HR were evaluated, and their values for the video game sessions and the 6 MWT were compared. The results were similar, but the jogging game was of greater intensity.
Holmes et al., 2013 [25]	A prospective, two-visit observational and cross-sectional study was conducted with 10 adults suffering from CF.	The intervention was first performed with the CPET for 15 min. Then, the participants played three video games on the Xbox Kinect console for 20 min.	VO2, CF, FVC, FEV1, SpO2, dyspnea, and fatigue were measured. Higher values were found for the CPET, but oxygen saturation was lower than that after playing on the Xbox.
Bingham et al., 2012 [27]	A crossover pilot clinical trial was performed with 13 adolescents with CF, who were randomly assigned to two groups that later crossed over.	One group used technical control software. The other group used feedback-based video games connected to a spirometer (15 min/day for 2–3 weeks).	The VC and FEV1 were measured.Higher values of the parameters were measured during the playing period than in the control periods.
Kuys et al., 2011 [21]	A randomized crossover trial was conducted with 19 adults hospitalized with CF. CG = 10 EG = 9	The EG participants played several video games for 15 min on the Nintendo Wii console. The CG participants used a stationary bicycle or treadmill for 15 min. The two groups crossed over later.	HR, SpO2, Kcal, MET, dyspnea, and fatigue were measured. Energy expenditures were similar for both groups, but the participants reported that the session with the Nintendo Wii was more satisfactory than the bicycle session.
Bingham et al., 2010 [26]	A pilot clinical trial was conducted with 10 children and adolescents hospitalized with CF.	In a computer video game, the participants inhaled and exhaled using a digital spirometer to keep a green ball on a path. Points were added or subtracted so that they could compete against themselves. Six sessions were held with 15 min per session.	Dyspnea, eye-breathing coordination, and the interest in using the spirometer in a play context were assessed. The results were positive, and how the participants controlled the respiratory flow without fatigue was recorded.

CG: Control group. EG: Experimental group. CF: Cystic Fibrosis. MET: Metabolic equivalent task. CPET: Cardiopulmonary exercise test. 6 MWT: Six-minute walk test. MSWT: Modified shuttle walk test. HJT: Horizontal jump test. MBTT: Medicine ball throwing test. HGT: Handgrip test. HR: Heart rate. SpO2: Oxygen saturation VO2: Oxygen consumption. VE: Lung ventilation per minute. Kcal: Kilocalories. FVC: Forced vital capacity. FEV1: Expiratory volume in the first second. VC: Vital capacity.

**Table 3 sensors-22-01902-t003:** The results of the methodological quality of the studies were determined using the PEDro scale.

Item (PEDro Scale)	1	2	3	4	5	6	7	8	9	10	11	Total Score
Del Corral et al., [18] 2018	X	X	X	X	N	N	X	X	X	X	X	8/10
Salonini et al., [19] 2015	X	X	X	X	N	N	N	X	X	X	X	7/10
Bingham et al., [26] 2012	N	X	N	X	N	N	X	N	N	X	X	5/10
Kuys et al., [20] 2011	X	X	X	X	N	N	X	X	X	X	X	8/10
Bingham et al., [25] 2010	N	X	N	X	N	N	X	N	N	X	X	5/10

N: the criterion is not satisfied; X: the criterion is satisfied. Note: Eligibility criteria items do not contribute to the total score. 1, Eligibility criteria; 2, Random allocation; 3, Concealed allocation; 4, Baseline comparability; 5, Blind subjects; 6, Blind therapists; 7, Blind assessors; 8, Adequate follow-up; 9, Intention-to-treat analysis; 10, Between-group comparisons; and 11, Point estimates and variability.

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
