# Peer review of "Effectiveness of Video Games as Physical Treatment in Patients with Cystic Fibrosis: Systematic Review"

_sensors, 2022, doi:10.3390/s22051902_

Round 1

Reviewer 1 Report

Introduction:

Why is this review necessary? How does it add to the what is already known?

Towards the bottom There is little research on CF and video games, but some studies suggest that they help gain strength and exercise tolerance, influencing quality of life and good adherence to home treatment [14,15]. I’d suggest change “they help” to suggests videos games enhance strength and exercise tolerance.

Intervention Section:

For study 22. What else did they find?

Author Response

Please, see the attachment. Thank you very much

Reviewer 2 Report

The paper presents a systematic review regarding the effectiveness of video games in patients with cystic fibrosis.

The review identified 809 records in various databases but in the end only 89 records were within the scope of the review. The review is well conducted however there are some aspects that may improve the quality of the paper:

  1. Please define every acronym at first use even if the acronym is well known in the literature (CFTR, 6MWT, MSWT, etc).
  2. The review refers to the effectiveness of video games in patients with cystic fibrosis however at the end of the review this is not discussed. Please provide data regarding the obtained effectiveness of the video games therapy as result of the study conducted.

Best regards

Author Response

Please see the attachment, thank you very much

Reviewer 3 Report

The review by López-Liria focuses on the influences of video game therapy on patients with Cystic Fibrosis. This manuscript reads well and is well developed and explained. Also, authors put on the table an alternative physical training at home for patients with chronic respiratory diseases.

I really enjoyed reading it and it gave me another perspective on respiratory health improvements through the use of video games that incorporate physical movement.

Two minor comments:

In my opinion, the title should be improved by highlighting the "physical movement" in it because it could be related at first sight to a sedentary lifestyle.

Likewise, it should be discussed the point that patients with CF have a higher risk of COVID and could combine their preventive isolation with this time of activities, in which there is a motivation to maintain and/or improve their health.

Author Response

(The authors gave the same response as above.)

Round 2

Reviewer 2 Report

I have no further comments.

Best regards.

PS. the guidelines for the authors are still in the text of the manuscript please delete them